Feeding behaviour of Caenorhabditis elegans is an indicator of Pseudomonas aeruginosa PAO1 virulence

Lewenza Shawn slewenza@ucalgary.ca
Charron-Mazenod Laetitia
Giroux Lauriane
Zamponi Alexandra D.
Department of Microbiology, Immunology and Infectious Diseases, Snyder Institute for Chronic Diseases, University of Calgary , Calgary, AB , Canada
Wiles Siouxsie
Electronic publication date: 2014 Aug 12
Publication date: 2014
Volume: 2
Electronic Location ID: e521
Received 2014 Mar 19; Accepted 2014 Jul 24
Copyright: © 2014 Lewenza et al.
Copyright year: 2014
Copyright holder: Lewenza et al.
License: This is an open access article distributed under the terms of the Creative Commons Attribution License, which permits unrestricted use, distribution, reproduction and adaptation in any medium and for any purpose provided that it is properly attributed. For attribution, the original author(s), title, publication source (PeerJ) and either DOI or URL of the article must be cited.
License URL: https://creativecommons.org/licenses/by/4.0/

Keywords: Caenorhabditis elegans, Pseudomonas aeruginosa, Type III secretion, Nematode feeding behaviour, High throughput virulence model, Food preferences

Funding: Cystic Fibrosis Canada This research was funded by Cystic Fibrosis Canada and received support from the Westaim-ASRA Chair in Biofilm Research, held by SL. The funders had no role in study design, data collection and analysis, decision to publish, or preparation of the manuscript.

==============================
Caenorhabditis elegans is commonly used as an infection model for pathogenesis studies in Pseudomonas aeruginosa. The standard virulence assays rely on the slow and fast killing or paralysis of nematodes but here we developed a behaviour assay to monitor the preferred bacterial food sources of C. elegans. We monitored the food preferences of nematodes fed the wild type PAO1 and mutants in the type III secretion (T3S) system, which is a conserved mechanism to inject secreted effectors into the host cell cytosol. A ΔexsEΔpscD mutant defective for type III secretion served as a preferred food source, while an ΔexsE mutant that overexpresses the T3S effectors was avoided. Both food sources were ingested and observed in the gastrointestinal tract. Using the slow killing assay, we showed that the ΔexsEΔpscD had reduced virulence and thus confirmed that preferred food sources are less virulent than the wild type. Next we developed a high throughput feeding behaviour assay with 48 possible food colonies in order to screen a transposon mutant library and identify potential virulence genes. C. elegans identified and consumed preferred food colonies from a grid of 48 choices. The mutants identified as preferred food sources included known virulence genes, as well as novel genes not identified in previous C. elegans infection studies. Slow killing assays were performed and confirmed that several preferred food sources also showed reduced virulence. We propose that C. elegans feeding behaviour can be used as a sensitive indicator of virulence for P. aeruginosa PAO1.

Introduction

C. elegans is an important model organism for developmental biology and infectious disease research. The nematode has been used for numerous studies as an infection host for Pseudomonas aeruginosa and many other bacteria (Sifri, Begun & Ausubel, 2005). When C. elegans is fed a lawn of P. aeruginosa PA14, the gut is colonized and death results within a period of days, also known as slow killing (Tan, Mahajan-Miklos & Ausubel, 1999; Feinbaum et al., 2012). In contrast, the fast killing assay results in C. elegans death within a period of hours. While the fast killing assay does not require live bacteria and is due to the production of secreted toxins including phenazines (Cezairliyan et al., 2013), the slow killing pathway requires ingestion of live bacteria and many different virulence genes (Feinbaum et al., 2012). P. aeruginosa has been shown to survive ingestion, colonize within an extracellular matrix in the nematode lumen, and cause much less damage to the intestinal epithelial barrier than S. aureus (Sifri, Begun & Ausubel, 2005; Irazoqui et al., 2010). P. aeruginosa strains display a range of virulence phenotypes, where PA14 is among the most virulent and PAO1 is among the strains with moderate slow killing activity (Lee et al., 2006). The virulence requirements may vary between P. aeruginosa isolates as PAO1 was shown to induce a rapid, paralytic killing mechanism dependent on hydrogen cyanide production (Gallagher & Manoil, 2001). In addition, P. aeruginosa also causes the red death phenotype after ingestion by C. elegans. In this infection model, P. aeruginosa is grown under phosphate limiting conditions, which leads to a red colored PQS-Fe3+ complex in dead nematodes and requires the pyoverdin acquisition pathway, the phosphate sensing PhoB response regulator and the MvfR-PQS quorum sensing regulator (Zaborin et al., 2009). The red death phenotype highlights the influence of growth conditions on virulence factor production and the mechanism of killing C. elegans.

Several high throughput methods have been developed to screen large transposon mutant libraries to identify virulence genes. Garvis et al. (2009) developed a high throughput screen of 2,200 transposon mutants for defects in C. elegans killing using a liquid assay and 24 h exposure to P. aeruginosa. Kirienko et al. (2013) also used liquid killing assays to show that P. aeruginosa requires the siderophore pyoverdin to kill C. elegans in liquid conditions. A genome-wide collection of transposon mutants in PA14 was tested for defects in C. elegans killing. The primary screen relied on identifying mutants that led to an increased brood size, and the virulence phenotype was confirmed in secondary screens using the slow killing assay (Feinbaum et al., 2012).

While killing assays measure the effect of single food sources on worm lethality or brood sizes, it is known that C. elegans has feeding preferences for certain species of bacteria (Zhang, Lu & Bargmann, 2005; Abada et al., 2009; Freyth et al., 2010). The choice index allows one to compare the preference of C. elegans for the conventional food source Escherichia coli OP50 to other bacterial food sources in binary assays (Zhang, Lu & Bargmann, 2005; Abada et al., 2009). Bacterial food sources that are preferred over E. coli OP50 have health benefits that include an increased life span and reproductive fitness of C. elegans (Abada et al., 2009; Freyth et al., 2010). If nematodes are raised on multiple food colonies that include E. coli OP50 and potential pathogenic species, C. elegans learns to avoid pathogenic isolates, in contrast to nematodes reared exclusively on the OP50 food source and then presented pathogenic food sources (Zhang, Lu & Bargmann, 2005). It is not surprising that C. elegans can identify potentially harmful pathogens as food sources given the nematode lifestyle of foraging for bacteria in rotting plants and soil (Schulenburg & Ewbank, 2007). C. elegans can be chemoattracted or repelled by certain diffusible bacterial odorants, which affects food choice selection (Bargmann, Hartwieg & Horvitz, 1993; Schulenburg & Ewbank, 2007). Bacillus nematocida acts as a nematode predator as it both attracts and kills C. elegans (Niu et al., 2010). Nematode feeding behaviour, sensing and decision-making regarding food sources is complex and involves multiple chemosensory pathways (Schulenburg & Ewbank, 2007).

The type III secretion system is a conserved secretion pathway in P. aeruginosa and many other pathogens, where effectors are secreted out of the bacterial cell and into the host cell cytoplasm through a needle complex (Hauser, 2009). The main three P. aeruginosa effectors (ExoY, ExoS, ExoT) are required to block wound healing and the macrophage inflammatory response, to kill phagocytes and disrupt epithelial barriers (Hauser, 2009). One report has shown that mutation of the exsA regulator of the type III secretion system is not required for P. aeruginosa PA14 killing of C. elegans (Wareham, Papakonstantinopoulou & Curtis, 2005). In this study, we revisited the role of the PAO1 type III secretion system in C. elegans virulence using a panel of strains that overexpress or were defective for type III secretion. We showed that a type III secretion deficient strain is a preferred food source and is impaired for virulence in the slow killing assay, suggesting a link between food preference and bacterial virulence. We screened a mini-Tn5-lux mutant library to identify mutants that served as preferred food sources and showed that these mutants have reduced virulence in the slow killing assay. Monitoring the nematode food source preference is a simple approach to identify potential virulence genes in Pseudomonas aeruginosa PAO1.

Materials and Methods

Strains and growth conditions

The Caenorhabditis elegans Bristol strain N2 was used for feeding and bacterial killing experiments (provided by Dr Kunyan Zhang). The tph-1 mutant worm is deficient in tryptophan hydroxylase required for the biosynthesis of serotonin (provided by Dr Jim McGhee). Escherichia coli OP50 was used as a non-pathogenic food source for cultivating C. elegans. The panel of type III secretion mutant strains and the wild type PAO1F (PAO1 from Alain Filloux) are described in Table 1 (provided by Dr Arne Rietsch). Wild type P. aeruginosa PAO1 and all mini-Tn5-lux transposon mutants (see Table 2) were previously described (Lewenza et al., 2005). Fifty µl of overnight OP50 cultures was spread and grown on NGM, which was used as a food source to cultivate nematodes. The NGM medium is composed of double distilled water, 0.25% (w/v) Bacto-Peptone (BD), 0.3% (w/v) NaCl, and 2% (w/v) Bacto-Agar (BD), 5 µg/ml cholesterol, 1 mM MgSO4, 25 mM KH2PO4 (pH 6) and 1 mM CaCl2. The SK medium is similar to NGM but contains 0.35% (w/v) Bacto-Peptone (BD).

Table 1 Type III secretion strains used in this study.

Strains	Description	Source	
PAO1F	Wild type PAO1 from
Alain Filloux’s lab		
ΔexsE	Loss of negative regulator
of type III secretion,
overexpresses exoY,
exoS and exoT	Cisz, Lee & Rietsch (2008)	
ΔexsEΔpscD	Mutation in type III
secretion machinery	A Rietsch, 2014, unpublished data	
ΔexsEΔexoSY T	Triple mutant ΔexoSY T
for type III effectors	Cisz, Lee & Rietsch (2008)	
ΔexsEΔexoTΔexoY (S+)	ExoS overexpression	Cisz, Lee & Rietsch (2008)	
ΔexsEΔexoSΔexoY (T+)	ExoT overexpression	Cisz, Lee & Rietsch (2008)	
ΔexsEΔexoSΔexoT(Y +)	ExoY overexpression	Cisz, Lee & Rietsch (2008)	
ΔexsEexoS :: GL3	exoS::gfp in exoS locus,
ΔexsE background	Cisz, Lee & Rietsch (2008)	

Table 2 Preferred food sources.

PAO1 transposon mutants that are preferred food sources to C. elegans.

Mutant ID	Insertion site	Gene	PA	Gene description	Screena	
11_B8	Intergenic	PA0120–PA0121		Transcriptional regulator–transcriptional regulator	B	
17_B11	Intergenic	PA4353–PA4354		Hypothetical-transcriptional regulator	B	
20_D11	Intergenic	PA0006–PA0007		Histidinol phosphatase-hypothetical	B	
11_B4	Gene	PA0056	PA0056	Probable transcriptional regulator	B	
12_G5	Gene	PA0578	PA0578	Conserved hypothetical protein	B	
16_E10	Gene	ksgA	PA0592	rRNA (adenine-N6,N6)-dimethyltransferase	B	
23_C9	Gene	agtS	PA0600	Two-component sensor	B	
83_C1	Gene	trpC	PA0651	Indole-3-glycerol-phosphate synthase	A, C	
52_D11	Gene	PA0667	PA0667	Putative metallopeptidase	B	
37_C7	Gene	PA0929	PA0929	Two-component response regulator	A, C	
52_B2	Gene	PA0930	PA0930	Two-component sensor	A, C	
23_D6	Gene	pqsB	PA0997	3-oxoacyl-[acyl-carrier-protein] synthase III	A	
32_D10	Gene	pqsD	PA0999	3-oxoacyl-[acyl-carrier-protein]	A, B, C	
76_C11	Gene	pqsE	PA1000	Quinolone signal response protein	A	
44_H6	Gene	mvfR	PA1003	Transcriptional regulator of PQS synthesis	B, C	
phoQ::xylE	Gene	phoQ	PA1180	Mg2+ sensing two-component sensor	B, C	
23_B7	Gene	PA1291	PA1291	Putative hydrolase	B	
11_F7	Gene	PA2906	PA2906	Probable oxidoreductase	B	
12_D9	Gene	cifR	PA2931	Cif transcriptional repressor	B	
12_H2	Gene	nqrB	PA2998	Na+-translocating NADH:ubiquinone oxidoreductase	A, C	
20_B2	Gene	mmsB	PA3569	3-hydroxyisobutyrate dehydrogenase	B	
69_A6	Gene	PA3747	PA3747	ABC-transport permease	B	
76_D11	Gene	tgt	PA3823	Queuine tRNA-ribosyltransferase	A, C	
50_D9	Gene	pprB	PA4296	Two-component response regulator	A, B, C	
17_B9	Gene	PA4497	PA4497	Binding protein component of ABC transporter	B	
19_D2	Gene	pilV	PA4551	Type 4 fimbrial biogenesis protein PilV	A	
11_E1	Gene	PA4714	PA4714	Predicted metal binding protein	B	
12_F5	Gene	PA4936	PA4936	Probable rRNA methylase	B	
26_C3	Gene	PA4983	PA4983	Two-component response regulator	A, C	
80_B7	Gene	aceF	PA5016	Dihydrolipoamide acetyltransferase	B	
47_B5	Gene	typA	PA5117	Regulatory GTPase	A	
18_H10	Gene	dctD	PA5166	Two-component response regulator	A, C	
12_B5	Gene	gcvT1	PA5215	Glycine-cleavage system protein T1	B	
68_G8	Gene	PA5228	PA5228	5-formyltetrahydrofolate cyclo-ligase	B	
52_F4	Gene	cbcX	PA5378	ABC-type choline transporter	B	
12_H1	Gene	PA5472	PA5472	ABC-type periplasmic transport protein	B	
14_D8	Gene	PA5498	PA5498	Probable adhesin	B	
Notes.

a Refers to method A or B used as a secondary screen to confirm the preferred food source phenotype in Fig. 3. C refers to mutants tested for virulence defects in the slow killing assay.

C. elegans feeding behaviour assays

Ten µl of normalized bacterial food sources (1 × 107 cfu/ml) were spotted on SK plates and grown overnight. Twenty L4 stage hermaphrodite nematodes were then transferred from E. coli OP50 NGM plates to the 6 cm assay SK plates containing the pre-grown P. aeruginosa food sources to be tested. SK plates with two, three or four P. aeruginosa food sources were monitored throughout a 72-h period. The number of worms in each bacterial colony was counted as an indicator of food preference. Within this time period, the original L4 worms could be easily differentiated from all new offspring on the basis of size. Feeding assays were performed at 25 °C. The unpaired t-test was used to compare the number of worms that selected mutant or wild type bacterial food sources at each time point. Images of colonies and worms were captured with a Motic DM143 digital microscope. When using tph-1 worms, the choice index was calculated using a previously described formula (choice index = test worms minus OP50 worms/total worms) (Zhang, Lu & Bargmann, 2005).

Fluorescence microscopy of C. elegans

L4 nematodes were given the choice of P. aeruginosa producing cherry fluorescent protein (ChFP) or green fluorescent protein (Gfp). The plasmid pCHAP6656 was introduced into wild type PAO1 as the Chfp-tagged strain, and exoS::GL3 ΔexsE was used as the Gfp-tagged strain. The pCHAP6656 plasmid expresses Chfp as outer membrane-localized lipoprotein (Lewenza, Mhlanga & Pugsley, 2008) and the plasmid did not affect the C. elegans response to PAO1. Nematodes were removed from either the Chfp or Gfp-tagged bacterial colonies and mounted on microscope slides for visualization. Fluorescent bacteria in the nematode gut were observed using a Leica DMI4000 B inverted microscope equipped with an ORCA R2 digital camera. The following excitation and emission filters were used to monitor red and green fluorescence, respectively (Ex 555/25; Em 605/52; Ex 490/20; Em 525/36). The Quorum Angstrom Optigrid (MetaMorph) acquisition software was used for image acquisition with a 63 × 1.4 objective and deconvolution was performed with Huygens Essential (Scientific Volume Imaging B.V.).

High throughput assay to identify preferred bacterial food sources

We previously constructed a mini-Tn5-lux transposon mutant library and mapped the transposon insertion site in 2,370 individual mutants (Lewenza et al., 2005). As a primary screen, this collection of transposon mutants was grown overnight in LB medium in 25 × 96 -well microplates. A 48-pin stamp was used to stamp transfer ∼5 µl of liquid LB culture onto a 6 × 8 grid of colonies on 15 cm SK agar plates and grown overnight at 37 °C. Ten L4 nematodes were deposited on each side of the grid (20 worms in total) and were allowed to eat and reproduce over the course of 7 days at 25 °C. Each plate with 48 different food sources was monitored daily for the first disappearance of specific colonies, which we identified as preferred food sources. In general, the first colonies were consumed by day 4. After identification of the preferred food source colonies, the nematodes would continue to reproduce and eat all the remaining bacterial colonies to completion. Mutants identified in the primary screen of 2,370 strains were retested in secondary screens to confirm the preferred food source phenotype. In method A, 6 mutants were arrayed in 6 specific positions within a 6 × 8 grid and surrounded by wild type PAO1. Alternatively for method B, individual mutants were positioned in 3 consistent well positions in the middle of a 6 × 8 grid of wild type PAO1 (see Fig. 3). Ten L4 worms were added to the both sides of the plates (20 worms in total), which were monitored daily to identify the first colonies eaten to completion.

Growth defect analysis of transposon mutants

All mini-Tn5-lux mutants that served as preferred food sources for C. elegans in the primary screen were tested for growth defects in LB and SK liquid media. Briefly, each strain was grown overnight in LB cultures and diluted 1/500 into 100 µl of fresh LB or SK liquid media in 96-well microplates. Cultures were grown and monitored for the OD600 values every 20 min throughout 18 h growth at 37 °C using the Wallac V ictor3 luminescence plate reader. Mutations that caused growth defects in SK growth medium were excluded from further analysis to rule out the possibility that weakly grown colonies might be preferred by the nematodes due to the low cell density.

Nematode chemotaxis assays

Bacterial cultures were grown overnight and 1 ml of supernatant was collected by centrifugation and filter sterilized. Twenty µl of culture supernatant was spotted at either end of 6 cm SK plates and 15 L4 nematodes were transferred to the center of SK plates. The plates were monitored every 15 min throughout a 2 h period to count the number of nematodes that moved into the dried spot of bacterial supernatant. Each supernatant was tested three times.

Slow killing assays

As previously described (Powell & Ausubel, 2008), 10 µl (1 × 107 cfu/ml) of bacterial cultures were transferred to 6 cm SK plates and spread to form a bacterial lawn. Thirty L4 stage nematodes were transferred from NGM-OP50 to SK plates containing the pre-grown bacterial lawn of P. aeruginosa and incubated at 25 °C. SK plates were prepared with 25 µg/ml of 5-fluoro-2′-deoxyuridine (FUdR), a eukaryote DNA synthesis inhibitor that prevents the growth of egg offspring during the experiment. E. coli OP50 was fed nematodes as a negative control strain with limited killing of C. elegans after 10 days. Nematodes were monitored daily under a dissection microscope to detect unresponsive and dead worms. Kaplan–Meier survival curves were plotted for triplicate experiments (n = 30) with a total of 90 nematodes. Significant differences in C. elegans survival were determined using the log-rank test (Graph Pad Prism) as previously described (Hasshoff et al., 2007; Feinbaum et al., 2012; Korgaonkar et al., 2013).

Results and Discussion

Ingestion and food preference in binary feeding assays withC. elegans

The type III secretion system is an important virulence factor in P. aeruginosa but was shown previously to not have a role in C. elegans virulence (Wareham, Papakonstantinopoulou & Curtis, 2005). Regulation of the type III secretion system involves a negative regulator called ExsE, and an ΔexsE mutant overexpresses and showed increased secretion of the type III effectors ExoT, ExoY and ExoS into the culture supernatant after growth in inducing, low calcium conditions (Rietsch et al., 2005). The ΔexsE mutant also demonstrated increased cytotoxicity to host cells, presumably due to the increased gene expression of all type III secreted effectors that were injected into host cells upon contact (Rietsch et al., 2005). We were interested to determine if the ΔexsE mutant that overexpresses the type III effectors had an effect on C. elegans feeding behaviour. Using a binary feeding assay where nematodes are given the choice of two bacterial food sources, L4 stage nematodes preferred wild type PAO1 to the ΔexsE strain (Fig. 1A). This result suggested that overexpression of type III effectors was detected and avoided by the nematode.

Figure 1 C. elegans ingests but ultimately avoids a P. aeruginosa ΔexsE mutant that overexpresses the type III secreted effectors.

(A) Feeding preference of worms given the choice of wild type PAO1 and an ΔexsE mutant that overexpresses the type III secreted effectors. Values shown are the mean of six independent experiments (n = 20) where the number of total worms equals 120. Asterisks indicate a significant difference between the mutant and wild type (∗∗∗p < 0.001; ∗∗p < 0.01). Worms were removed from bacterial food source colonies and monitored for Chfp or Gfp-labelled bacteria in the GI tract. (B) At 6 h, worms selected from PAO1 (pCHAP6656) colonies or (C) worms selected from ΔexsE exoS-gfp colonies. (D) At 48 h, Gfp and Chfp-labelled bacteria in the GI tract. (E) A worm selected from outside the colonies. White arrows point to the GI tract. Scale bar, 15 µM.

To determine if the nematodes ingested both possible food sources, C. elegans was fed a Chfp-tagged wild type PAO1 (pCHAP6656) or Gfp-tagged, ΔexsE exoS-gfp strain and nematode intestinal tracts were monitored with microscopy. The ΔexsE exoS-gfp strain overexpresses exoY and exoT and was also avoided by nematodes, comparable to the ΔexsE strain that overexpressed exoY, exoT and exoS (data not shown). Although the worm body was autofluorescent in both green and red channels, the early stages of bacterial accumulation were observed (Fig. 1). At early time points, worms that were removed from wild type PAO1 (pCHAP6656) colonies were shown to have exclusively red fluorescent bacteria in the gastrointestinal tract (Fig. 1B). Similarly, worms taken at early time points from the ΔexsE exoS-gfp strain had exclusively green fluorescent bacteria in the GI tract (Fig. 1C). At later time points, worms were shown to have both green and red fluorescent bacteria in the GI tract (Fig. 1D). This indicates that worms ingested both food sources and did not simply avoid feeding on the ΔexsE strain. Some nematodes chose neither of the two food sources but as the exposure time increased, more worms made a food choice (data not shown). However, there were still worms that avoided both food sources and did not have any bacteria in the GI tract (1E). We concluded that the nematodes ingested and compared both possible food sources, but ultimately preferred the wild type strain.

Type III secretion mutants are preferred food sources and impaired for virulence

We reasoned that if type III overexpression strains were avoided, than secretion defective strains might be preferred. Indeed, C. elegans preferred strains that were defective in the type III secretion machinery (ΔexsEΔpscD) and were defective for all type III secretion effectors (ΔexoSY T) (Fig. 2A). After longer periods of feeding (∼4 days), the ΔexsEΔpscD food source was the most preferred as it was the first colony consumed (Fig. 2C). To compare the potential toxicity of type III effectors, nematodes were exposed to strains that overexpress individual effectors. In these feeding assays, nematodes consistently preferred the ExoY overexpressing strain suggesting ExoY is the least toxic effector (Fig. 2B). ExoY is an adenylate cyclase and our findings are comparable with a report showing that ExoY was not required for in vitro cytotoxicity and had no impact on dissemination during infection (Lee et al., 2005). To determine if the type III secretion system of PAO1 was required for C. elegans virulence, we performed slow killing assays with individual P. aeruginosa food sources. The ΔexsEΔpscD showed a significant reduction in slow killing of C. elegans, confirming that the most preferred food sources also show reduced virulence (Fig. 2D).

Figure 2 Type III secretion defective mutants are preferred food sources and less virulent in C. elegans slow killing.

(A) Feeding preference of worms given the choice of an ΔexsE mutant, a secretion defective ΔexsEΔpscD mutant and a triple effector mutant ΔexoSY T. Values shown are the mean of three independent experiments (n = 20) where the number of total worms equals 60. Asterisks indicate a significant difference between the secretion mutant and ΔexsE strain (∗∗p < 0.01). (B) Feeding preference of worms given the choice of wild type PAO1F and strains that overexpress either exoS+, exoY+ or exoT+. Values shown are the mean of three independent experiments (n = 20) where the number of total worms equals 60. Asterisks indicate a significant difference between the preferred strain and wild type (∗∗p < 0.01). (C) The ΔexsEΔpscD mutant was the first colony eaten to completion. (D). Kaplan–Meier survival curves for nematodes fed mutants in the type III secretion system. The % survival values represent three independent experiments (n = 30) where the total number of worms equals 90. E. coli OP50 is the non-pathogenic food source. Asterisks indicate a significant difference from the wild type PAO1F as determined by the log-rank test (∗∗∗p < 0.0001, ∗∗p < 0.01).

Figure 3 Preferred feeding behaviour in high throughput assays with a choice of 48 food sources for C. elegans.

The preferred food source phenotype in (A) method A, where six different mutants were grown at specific positions within a 48-grid of wild type PAO1 colonies. Solid circles indicate preferred food sources and dashed circles highlight partially eaten or uneaten colonies. (B) In method B, individual mutants were grown in triplicate at consistent positions within a 48-grid of wild type PAO1 colonies. As positive controls, heat killed PAO1 or E. coli OP50 were spotted in triplicate. The black rectangles highlight the triplicate positions of preferred food sources, which include transposon insertion mutants in pqsD, PA0667, mvfR or cbcX. All images were captured after 4 ± 1 day after the addition of 20 L4 nematodes.

High throughput feeding behaviour assays

Given the relationship between feeding preferences and virulence observed with strains defective for type III secretion, we wanted to screen a large collection of mutants to find additional preferred food sources. We previously constructed a large library of mini-Tn5-lux mutants with mapped insertion sites, which were arrayed into a library of 2,370 mutants in 96-well microplates (Lewenza et al., 2005). Using standard petri dishes with SK agar, we stamped liquid LB cultures onto a grid of 48 colonies (6 × 8), and introduced 20 L4 stage nematodes. The plates were incubated at 25 °C and observed daily to identify colonies that were preferentially eaten to completion. In the primary screen of 2,370 mutants, we identified 191 strains that were preferred food sources. Growth curves were performed on all candidates from the primary screen and those mutants with growth defects in SK medium were excluded from further study. We developed secondary screens to confirm this phenotype. In method A, we arranged 6 unique preferred food sources within a grid of 48 colonies, with PAO1 in all other positions (Fig. 3A). This method did confirm the preferred feeding source phenotype of some mutants. Since the worm was given a choice of multiple preferred food sources, there may have been competition for the most preferred. To reduce the competition observed in method A, we used an alternative secondary screen where individual candidates were situated in triplicate positions within a 48-grid of wild type PAO1 (Fig. 3B). Using these two secondary screens, we confirmed the preferred food source phenotype of 37 mutants (Table 2). After sufficient incubation time, the nematodes would reproduce to high numbers and eat all the bacterial colonies to completion.

Preferred food sources for C. elegans included mutants in many known virulence factors

Table 2 summarizes the transposon insertion sites of genes that led to the preferred food source status for C. elegans. The genes identified in this screen can be grouped into the following categories: known virulence factors, virulence regulatory systems, nutrient utilization and metabolism, and hypothetical proteins. The majority of the genes identified in PAO1 as having a preferred food source phenotype (Table 2) are present in PA14 but were not identified in the genome-wide screen for slow killing determinants in PA14 (Feinbaum et al., 2012). This indicates that the two assays are unique and identify different bacterial phenotypes. However, mutations in the pqs biosynthesis and type IV pili genes resulted in slow killing defects (Feinbaum et al., 2012) and led to preferred food source status (Table 2), indicating a small overlap between the two screens.

Mutations in genes encoding the PQS biosynthesis genes, type IV pili and the TypA GTPase had a preferred food source phenotype, and all were previously recognized as virulence factors in P. aeruginosa (Zaborin et al., 2009; Feinbaum et al., 2012; Neidig et al., 2013). Among the known regulators of virulence, we identified the PprB, PhoQ, PqsR and CifR regulatory systems (Cao et al., 2001; MacEachran, Stanton & O’Toole, 2008; Gooderham et al., 2009; de Bentzmann et al., 2012). The C. elegans virulence screens frequently identify global regulators and two-component system regulators, probably due to the pleiotropic effects of these mutations, given the large number of virulence genes controlled by these systems (Sifri, Begun & Ausubel, 2005). Both PqsR and PprB are known regulators of the pqsABCDE biosynthesis genes (Cao et al., 2001; de Bentzmann et al., 2012), which were previously shown to be involved in the C. elegans red death and slow killing phenotypes (Zaborin et al., 2009; Feinbaum et al., 2012). PhoQ is a two-component sensor that responds to limiting Mg2+ and controls numerous genes including antimicrobial peptide resistance modification to LPS (Macfarlane et al., 1999), and is required for virulence in plant and chronic rat lung infections (Gooderham et al., 2009). The CifR repressor controls the Cif secreted toxin that reduces the apical expression of CFTR and chloride secretion in epithelial cells (MacEachran et al., 2007; MacEachran, Stanton & O’Toole, 2008). PA14, but not PAO1, expresses Cif activity (Swiatecka-Urban et al., 2006), suggesting that the CifR mutant phenotype in this assay is independent of Cif activity.

Consistent with the identification of regulatory proteins in C. elegans virulence screens, we also identified mutations in several transcriptional regulators and two-component sensors and response regulators (PA0056, PA0929–PA0930, PA4983) that led a preferred food source phenotype (Table 2). Intergenic insertions between PA0120 and PA0121 likely disrupted the downstream PA0121, an uncharacterized transcriptional regulator. It is unclear what gene is affected by an intergenic insertion between PA4353–PA4354, due to their divergent orientation. The functions of PA0056, PA4983 and PA0929–PA0930 are currently unknown, and the latter two-component system is adjacent to gacS, a global virulence regulator required for C. elegans killing (Sifri, Begun & Ausubel, 2005).

Preferred food sources for C. elegans included mutants in nutrient acquisition pathways

This screen also identified mutants with insertions into pathways required for the uptake and utilization of various nutrient sources. The DctD two-component response regulator controls expression of the ABC transporters that take up C-4 dicarboxylates (Valentini, Storelli & Lapouge, 2011), and a mutation in this gene led to a preferred food source for C. elegans (Table 2). There were also mutations in multiple ABC transporters that led to preferred food source status, including transporters involved in the uptake of dipeptides, quaternary ammonium compounds (QAC), zinc, as well as genes involved in amino acid metabolism (Table 2). The AgtS (PA0600) two-component sensor regulates an ABC transporter that is required for the uptake of δ-aminovalerate and γ-aminobutyrate (Chou, Li & Lu, 2014). The cognate response regulator AgtR is also required to sense N-acetylglucosamine shed from gram-positive bacterial cell wall, leading to increased pyocyanin production and enhanced fruit fly killing during mixed infection (Korgaonkar et al., 2013). Mutations in this two-component system may affect both nutrient acquisition and pyocyanin production, which is directly toxic to C. elegans (Cezairliyan et al., 2013). These results are consistent with a previous screen that also identified various metabolic and nutrient acquisition genes required for slow killing of C. elegans by PA14 (Feinbaum et al., 2012).

Preferred food sources do not cause universal chemoattraction of C. elegans

A simple explanation for the preferred feeding phenotype is that bacterial mutants may have had increased production of an attractant or lacked production of a repellant that caused increased nematode accumulation in a colony. To test this hypothesis, we compared the nematode attraction to filtered culture supernatants from wild type PAO1 and preferred food sources in a binary chemoattraction assay. Culture supernatants were collected and spotted on SK plates at equal distances from where 15 L4 worms were transferred (Fig. 4A). Within 30 min, worms were more attracted to the culture supernatants of E. coli OP50 than PAO1 (Fig. 4B), which was also a preferred food source (Fig. 3). Of the supernatants tested, nematodes were more chemoattracted to supernatants from the PA4497 and PA1291 mutants (Fig. 4). In contrast, nematodes were equally attracted to the parent and mutant supernatant in the other combinations tested (Fig. 4). In one instance, C. elegans was repelled by one culture supernatant after 30 min (Fig. 4E). While it appeared that some supernatants might have contained a chemoattractant for C. elegans, no chemoattraction was observed in most cases.

Figure 4 Binary chemotaxis assays of C. elegans towards the culture supernatants from PAO1 and preferred food sources.

(A) Filtered culture supernatants from the wild type PAO1 and a preferred food source were spotted onto SK plates. Fifteen nematodes were transferred to the middle and their migration to either side of the dashed line was monitored. Binary assays compared nematode attraction between (B) PAO1 and E. coli OP50, or between PAO1 and transposon mutants in (C) PA4497, (D) PA1291, (E) PA0006–PA0007 (F) PA4353–PA4354 (G) mvfR and (H) PA2931. Values shown are the mean of three independent experiments (n = 15) where the total number of worms equals 45. Asterisks indicate a significant difference between the mutant strain and wild type (∗p < 0.1).

tph-1 worms demonstrated preferred feeding behaviour

The tph-1 nematode strain is defective for the enzyme tryptophan hydroxylase, which is the rate-limiting step in the biosynthesis of serotonin. This neurotransmitter was previously shown to be important for aversive learning and avoidance of P. aeruginosa (Zhang, Lu & Bargmann, 2005). Naive worms have a preference for P. aeruginosa over E. coli OP50, but learn to avoid virulent P. aeruginosa strains in binary choice assays if they were pre-exposed to P. aeruginosa for 4 h prior to the food choice experiment (Zhang, Lu & Bargmann, 2005). The avoidance behaviour is specific to the strains that were pre-exposed to the worms, and requires the serotonin neurotransmitter pathway (Zhang, Lu & Bargmann, 2005). Here we tested tph-1 worms for their preference of either the type III overexpression ΔexsE strain of P. aeruginosa or E. coli OP50. At an early time point (5 h), wild type N2 worms preferred P. aeruginosa ΔexsE, but preferred OP50 between 24 and 48 h exposure (Fig. 5A). The tph-1 mutant strain did not have any preference at 5 h, but also demonstrated a similar preference for OP50 between 24 and 48 h (Fig. 5A).

Figure 5 C. elegans tph-1 mutants are capable of avoidance and preferred feeding behaviours.

(A) Wild type N2 and tph-1 nematodes were given the choice of E. coli OP50 and an ΔexsE mutant and the choice index was determined throughout 48 h. Values shown are the averages and SEM of three independent experiments (n = 20) where the number of total worms equals 60. (B) The tph-1 nematodes were tested in feeding assays where mutants were embedded in triplicate within a 48-grid of wild type PAO1 colonies. The black rectangles highlight the position of preferred food sources, which included mutants in phoQ, PA0667, PA0592 and PA3747. All images were captured after 4 ± 1 day after the addition of 20 nematodes.

Given the rapid, aversive learning behaviour of C. elegans after only 4 h of pathogen pre-exposure (Zhang, Lu & Bargmann, 2005), we wanted to determine if the tph-1 worms were still capable of detecting preferred P. aeruginosa food sources over the time course of a few days in the feeding assay with a grid of 48 colonies. We tested a subset of the N2 preferred food sources and showed that tph-1 worms were still capable of detecting and eating those preferred sources to completion before moving on to eat the wild type PAO1 (Fig. 5B). This indicated that the serotonin chemosensory pathway was not required for C. elegans to differentiate the preferred food sources from the wild type PAO1 colonies that occupied the majority of positions in the food grid.

Preferred food sources were defective for full virulence inC. elegans slow killing assays

It is known that preferred bacterial food sources are associated with health benefits such as increased C. elegans life span and reproductive fitness (Abada et al., 2009; Freyth et al., 2010). We wanted to further establish a relationship between the preferred food status and virulence in C. elegans. We tested a panel of 11 mutants identified as preferred food sources (Table 2) for defects in slow killing of C. elegans. Interestingly, 45% of the preferred food sources tested showed significant defects in slow killing relative to the wild type PAO1 (Fig. 6). A pqsD mutant in the PQS biosynthesis pathway showed the strongest virulence defect, comparable to the non-pathogenic E. coli OP50 food source (Fig. 6). The pqs operon is required for the synthesis of the Pseudomonas quinolone signal, a quorum sensing signal molecule that contributes to slow killing and the red death phenotype in C. elegans (Zaborin et al., 2009; Feinbaum et al., 2012). PA4296 (PprB) is a two-component response regulator that regulates several adhesins, pili and fimbriae, the pqs operon, and is attenuated in the Drosophila acute and chronic infection models (de Bentzmann et al., 2012). PA0929–PA0930 encode a two-component system that has no known function, but is required for full virulence in C. elegans (Fig. 6). Lastly, PA4983 also encodes a two-component response regulator with no known function. These regulatory systems are excellent candidates as novel important regulators of virulence of C. elegans.

Figure 6 Preferred P. aeruginosa food sources have reduced virulence in the slow killing of C. elegans.

Kaplan–Meier survival curves for nematodes fed preferred food sources. The % survival values represent three independent experiments (n = 30) where the total number of worms equals 90. (A) E. coli OP50 is the avirulent food source control. Mutants in the genes encoding (B) pqsD, (C) PA4296, (D) PA0929, (E) PA0930 and (F) PA4983 show significant differences from the wild type PAO1 as determined by the log-rank test (∗∗∗p < 0.0001, ∗p < 0.05).

Conclusions

We describe a method to identify preferred bacterial food sources of C. elegans and present several lines of evidence to support this approach as a new strategy to identify P. aeruginosa virulence factors. We first used the feeding behaviour assay to demonstrate that the PAO1 type III secretion defective mutants were preferred food sources and also were impaired in the slow killing virulence assay. Secondly, preferred food sources were defective in several known virulence factors of P. aeruginosa, some of which were specifically shown to contribute to C. elegans killing. Nematode feeding preferences were not due simply to increased chemoattraction to secreted bacterial products. Given the diversity of genes identified in this screen, it is unlikely that these mutations affect conserved biosynthetic pathways of compounds that act as odorants and influence nematode chemoattraction. Lastly, we report that preferred food sources have a defect in virulence, or a beneficial outcome on C. elegans survival in the slow killing infection model. This approach has led to the discovery of many novel genes likely required for virulence, including multiple two-component sensor systems whose target genes are unknown.

It is interesting to note that comparisons between this study and other reports of P. aeruginosa virulence factors in C. elegans reveal little overlap in the bacterial requirements for worm killing (Tan, Mahajan-Miklos & Ausubel, 1999; Gallagher & Manoil, 2001; Garvis et al., 2009; Feinbaum et al., 2012; Kirienko et al., 2013; Zaborin et al., 2009). To date, these differences in P. aeruginosa virulence in the C. elegans infection model have been attributed to genetic or regulatory variation in bacterial strains, or the influence of growth conditions on bacterial virulence phenotypes. Not all of the preferred food sources tested had significant differences in nematode survival. Despite this observation, we propose that nematode feeding preferences is a more subtle indicator of virulence than slow killing. This simple method to identify potential bacterial virulence factors for C. elegans should be applicable to other pathogens. The precise function and contribution to virulence will require further study. Recently, a similar high throughput assay was used to screen transposon mutants of P. fluorescens NZI7 to identify the repellants that deter grazing by C. elegans (Burlinson et al., 2013).

Further research is needed to understand the mechanisms to account for the nematode food preferences. Pathogen avoidance mechanisms are studied in C. elegans, which act as behavioral defenses against pathogenic microbes (Schulenburg & Ewbank, 2007). The first step in pathogen detection may involve mechanosensory or chemosensory discrimination. C. elegans avoids Serratia marcescens using G protein-like receptors, AWB chemosensory neurons and the Toll-like receptor gene tol-1, and mutations in these pathways reduce the capacity of C. elegans to discriminate strains producing a secreted bacterial surfactant, serrawettin W2 (Schulenburg & Ewbank, 2007). Nematodes can remember exposure to pathogenic species like P. aeruginosa and then avoid these food sources using the serotonin signalling pathway (Zhang, Lu & Bargmann, 2005), however, we have shown that tph-1 mutants can still discriminate preferred food sources. In C. elegans, an insulin-like signalling pathway is required to respond to general environmental stresses and to physically evade pathogenic Bacillus thuringiensis species (Hasshoff et al., 2007). Feeding behaviour assays are a useful approach to studying bacterial pathogenesis and highlight the nematode’s ability to make decisions and identify less virulent or edible food sources.

The authors acknowledge JoAnn McClure and Dr Kaiyu Wu for assistance with worm handling and methods, Dr Jim McGhee for assistance with worm methods and providing the tph-1 strain, and Dr Arne Rietsch for providing the panel of type III secretion mutant strains.

Additional Information and Declarations

Competing Interests

Author Contributions

The authors declare there are no competing interests.

Shawn Lewenza conceived and designed the experiments, analyzed the data, contributed reagents/materials/analysis tools, wrote the paper, prepared figures and/or tables, reviewed drafts of the paper.

Laetitia Charron-Mazenod conceived and designed the experiments, performed the experiments, analyzed the data, wrote the paper, prepared figures and/or tables, reviewed drafts of the paper.

Lauriane Giroux conceived and designed the experiments, performed the experiments, analyzed the data, reviewed drafts of the paper.

Alexandra D. Zamponi performed the experiments, analyzed the data, reviewed drafts of the paper.

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
