# Peer review of "Feeding behaviour of Caenorhabditis elegans is an indicator of Pseudomonas aeruginosa PAO1 virulence"

_PeerJ, doi:10.7717/peerj.521_

## Round 0.1 · original submission · Major Revisions

As raised by Reviewer 2, I have reservations about the experimental design used. In addition to their comments, can you please indicate how many times the experiments shown in Figures 4 and 6 were carried out. It would be easier to compare the data in Figure 6 if the area under curve (AUC) values were presented instead of the current line graphs. Data presented as bar graphs of averages and SEM would be more appropriately presented as the individual data points.

Reviewer 1 ·

Basic reporting

1. The abstract should emphasize their findings that additional genes in PA01 were identified using the food preference assay that were not previously characterized for their role in virulence in killing C. elegans.

2. In the introduction section, line 24-25, the statement “C. elegans is a bacteriovore that forages for bacteria in rotting plants and soil” seems a little out of context between the previous sentence and the next sentence. Please rephrase to tie in the natural food behaviour demonstrated in its natural habitat to avoid potentially harmful pathogens or take this statement out completely and address it in the discussion section to strengthen the link with the serotonin neurotransmitter.

3. In the methods section, line 102 – the reference given for the explanation of pCHAP6619 lacked details on this plasmid. Please provide the details.

4. Line 161, reference the figure after the observation statement; in this case, Figure 1A.

5. Line 174, the phrase “…chose one of the two food sources …” should be replaced with “…chose one of the food sources at the timepoints monitored”. The original statement infers that the worms were monitored constantly when in fact the worms were only monitored at the defined timepoints.

6. The tph-1 mutant strain still displayed food preference behaviour; have the authors considered trying the tol-1(nr2033) mutant worm as reported elsewhere for deficient olfactory sensing? If not, it may be worthwhile to mention this in the discussion.

Experimental design

1. Regarding data presented in Figure 1A, Figure 2A, and figure 2B, was the total number of the worms at the end of the assay after 5 days the same as the starting total number of worms? Were any original subject worms that died during the assay censured from analysis? Were progeny worms included in the totals or were they removed from the feeding behaviour assay plates? In accordance to Figure 6, approximately 50% of the worm population remained after 5 days on PA01 – was this taken into account the survival of worms in the feeding behaviour assay plates in Figure 1A? This information should be added to the method and/or results sections.

2. Figure 1B and 1C, it is difficult to distinguish between fluorescent disintegrated bacteria and fluorescent intact bacteria based on the fluorescence and scale in the micrographs. Is there any way to improve the image such that intact fluorescent bacteria may be observed? Also, I would disagree with the use of the term “distended” – a cited paper (Tan et al., 1999) reported that intestinal distension was not observed until at least 24- 48 hours post infection. Because the timepoint of 6 hours was chosen, the phrase “initial stages of bacterial accumulation were observed” might be better suited. Further to this, were the timelines for bacterial accumulation approximately the same or different between PA01 and exsE? This could be an interesting observation to be added to the results section.

3. The manuscript made reference to a previous study (Wareham, Papakonstantinopoulou and Curtis, 2005) indicating that the type III secretion system did not appear to be a requirement for killing of C. elegans. Given that PA01 is used in this study in comparison to PA014 used in the previous study, did the authors do C. elegans killing assays to confirm that the case is the same for PA01, ΔexsE, ΔexsEΔpscD as well as hypersecretion strains for ExoS, ExoY and ExoT? This is particularly important especially citing differing virulence phenotypes due to strain variation in the discussion section.

4. Regarding the killing assay data shown in Figure 6, statistical analyses should be included to indicate significant differences, if any, to support the conclusions stated in lines 306-307. Ideally, the data should be displayed in Kaplan-Meier survival curve format supported by log-rank analyses as commonly done in the publications concerning C. elegans killing assays. In addition, a statement should be made in the methods section regarding the number of independent experiments were done to assess the virulent phenotype of the strains concerned. Generally, at least three independent experiments are done with the graph of one experiment used as a representative graph to be placed in a figure.

Validity of the findings

The manuscript submitted by Lewenza and colleagues describes a new screening assay for virulence factors based on C. elegans feeding behaviour. The authors propose that the screening assay is more sensitive than conventional killing assays and has led to the identification of “known” virulence genes that have not been yet characterized in killing of C. elegans. In addition, this assay is amenable to high-throughput screening. The manuscript touches on the ability of C. elegans to "smell" degrees of pathogenicity - something that is often overlooked in C. elegans killing assays. Overall, the manuscript is well-written and the data is sound though clarification is needed on the methodologies used and presentation of the data. The discussion is a little on the short side; the authors may want to expand on the link between olfactory sensing and virulence phenotypes.

Daniele Provenzano ·

Basic reporting

Abstract:
The abstract of this manuscript should be rewritten in its entirety because it is very hard to understand, devoid of transitions and riddled with gaps and inconsistencies. For example, the second sentence describing the use of nematodes in assaying bacterial virulence finishing at line 5 is disconnected from the following sentence that introduces Type III secretion system role (or lack thereof) in worm killing. The terminology “competent in hyper secretion” is employed throughout the manuscript, although surely not wrong, does not facilitate comprehension of the text, I suggest substituting this with “hypersecreting” or “overexpressing” throughout the text. Clearly, in line 9 the authors meant to write C. elegans instead of PAO1. The abstract states that the 37 mutants identified in the screen from the transponson mutant library included those of mutants in virulence genes; while the wording is correct it is also loose and misleading, as numerous other mutant strains were also selected. This needs to be sorted out and described accurately. The suggestion that the feeding assays employed in this work can be used as sensitive indicators of virulence is inconsistent with the data shown and should be removed. In the abstract food sources (although implicit) are not defined and neither is the binary feeding assay. Only by reading the body of the article does the meaning of the abstract becomes clear; this is not the purpose of the abstract, it should summarize the body of work independently of the entire article.

Introduction:
Line 26: P. aeruginosa strain PA14 is not introduced hence, the contextual meaning of the statement remains elusive.
Line 28: Define “synchronized worms”.
Line 30: Define ”biofilm-like extracellular matrix” as biofilm components produced by P. aeruginosa are very well studied.
Line 31: Define “to what extent” epithelial cells were invaded, if this is relevant or remove it.
Lines 32: Define and explain the “slow killing assay”.
Lines 34-38: Apparently “the fast killing pathway” is mediated solely by the secreted phenazine secreted by strain PA14, because apparently PAO1 does not secrete phenazine and therefore cannot employ the “fast killing pathway” as worded by the authors. Yet PAO1 can induce another form of rapid killing mediated by cyanide production? This illustrates that the definitions of terms are unclear. Ether killing is fast or slow; speed of killing is not dependent on which factor carries out the lethal event, but by definition, on speed. I encourage the authors to sort this out.
Lines 46: Define “red death killing”.
Line 53: introduce and define PO50, as it isn’t even clear to the reader that this is E. coli.
Line 54: After the abstract, this is the first mention of binary feeding assay. Clearly the term obviously suggests that the worms are being offered two sources of food, but it would be appropriate for the authors to explain this and how this is carried out here.
Lines 59 through 63 complicate rather than simplify comprehension and since they are, at this point not relevant, I suggest removing them. Instead, it might be useful to introduce life cycle stages of worms and a description of them here as they are referenced later in the manuscript. For example, worm size, sexual cycle, life span, if any, or whatever else may be helpful to the reader to understand the biology of the worms in the life cycle stages employed in this model should be described in the introduction briefly, clearly and succinctly.
Line 65: a brief paragraph about Type III Secretion System (T3SS) might also be useful since this is clearly an important virulence factor in other models, but seemingly less in C. elegans.
Line 70-73: The sentence that begins with “Based on this subtle worm feeding behavior ….” is grammatically incorrect and unintelligible and needs to be entirely reworded.

Conclusions:
Line 325: the authors claim their results hardly reproduce those of others, this is not encouraging.
Lines 330-334: this may be a conclusion that could be drawn from other investigators work, but from this report.

Experimental design

Materials and Methods:

Positive and negative controls should be described in each section of the Material and Methods.
Line 77: I am aware that C. elegans has a hermaphroditic stage this should be mentioned, but are these worms clonal? How are they propagated to insure and isogenic offspring for experiments?
Line 92: Define L4 worms as requested previously in comments pertaining to the introduction.
Lines 89-99: The paragraph following the heading “C. elegans feeding behavior assays” must be the section where the “binary feeding assay” referred to repeatedly throughout the paper is described; I encourage the authors to state this very clearly. Apparently all that entails is placing worms on agar plates prespotted with O/N grown bacteria in spots. No assurance is being provided that the distance between the worm inoculums and the bacterial outgrowths is the same, that this even matters is not addressed, how the bacterial spots are oriented on the plates relative to the worms, that the distance the worms have to travel to reach each spot is equal. Considering the critical nature the “binary feeding assay” should be described very clearly and unambiguously. Even though this model was developed and has been used by others previously it would be useful for the reader to know how this assay was carried out by the authors of this manuscript for reproducibility’s sake.
Lines 101-105: Were controls carried out by swapping the fluorescent marker between the mutants employed to insure the reader that expression of fluorescent markers did not affect the feeding preference of the worms? Furthermore, it appears that the two strain expressing fluorescent proteins display additional genotypic differences, the exoS deletion strain expresses gfp chromosomally and the wild type rfp episomally, I encourage the authors to reassure the readers that this has no effect on worm behavior (this difference should have been controlled at some point).
Line 102: pCHAP6619 appears to be a plasmid, a plasmid can be harbored but the genes on it are expressed, not the plasmid itself. Please correct this inaccuracy; it is confusing to student readers.
Lines 119-120: How long does it take for worms to reproduce to high numbers and eat all the colonies regardless of genotype to completion? This would be useful to know relative to the 3-5 day monitoring described. Authors surely must have used a single cut-off incubation time to present the images shown in figure 5, otherwise the selection would be based on arbitrary choices and therefore biased.
Lines 120-126: The authors describe the preferential feeding assay employed to screen the transposon mutant library. However, the description is too vague to reproduce the experiments as the two screening methods are not distinguished clearly as screen A and screen B as indicated in Table 2. Once again, the authors should state precisely when the decision was made to determine whether a mutant was preferentially fed on or not, the description between day 3 and 5 without qualification would otherwise be biased. Was the screen conducted in PAO1 or in PAO1F?
Lines 128-133: The heading “Growth defect analysis” describes how mutants that displayed growth defects were excluded from further consideration (something that isn’t sufficiently underscored in the results section). The authors should state if they did this after each screen or at the conclusion of both screens. Regardless, I encourage the authors to repeat these experiments in M9 minimal media as this would clearly narrow the pool of candidate mutants further greatly and improve their screen.
Line 137: change to: “… from the location where fifteen L4 worms were deposited on SK plates.”
Table 1 is incomplete, it does not describe the background of mutant strains (it is assumed that this should be PAO1F for all), but reassurance of this in “strains and growth conditions” section on lines 77-87 or in table 1 would be helpful. The strains in the slow killing assay (Figure 6) are not listed/described in table 1. Why is PAO1 from Alain Filloux’s lab referred to by an additional F? Is it different than the PAO1 strain other labs use?
Lastly, since much of this data is quantitative, a section on how statistical analyses were carried out would be an appropriate addition to the Methods and Materials section. Please introduce SEM acronym (probably Standard Error of Means, but nevertheless).

Results and Discussion:
Lines 150-157 are introductory and should be moved to the introduction where strain phenotypes, the binary feeding assay and the L4 stage worms should be introduced, as suggested previously.
Figure 1A: What does “None” precisely mean? If this means that this is the number of worms that is not “near” either bacterial spots, how was this determined? What is the cut-off distance that was used to determine whether a worm is sufficiently near any bacterial spot to qualify it as feeding on it vs. not and how was this distance ascertained?
Lines 165-167: Seems evident that worms that after 2 hours had fed exclusively on either GFP or RFP expressing bacteria had green and red bacteria in their gut respectively.
Line 169: what section of Figure 1 is being referenced? Parts D and E?
Line 182: use nomenclature in the text that rapidly identifies strains in the figures across the manuscript consistently (delta3TOX).
Figure 2C contains no reference to size. The figure is very hard to interpret for anyone not knowing what the images represent because the bacteria are spotted and are not uniform CFUs and the worms measure 1 mm.
Data shown in figure 1A becomes statistically significant only after 48 hours, figure 2A after 72 hours but figure 2B after 24. These wide differences in time laps to obtain statistically significant results in their assay are not addressed by the authors but should be.
Line 183: correct deltapscD to deltaesxE/deltapscD.
Lines 191 onwards: No mention is made to the growth curves performed on mutants to narrow the pool of the screen in an attempt to assess growth defects as described in the Methods and Materials section and how these growth curves were used.
Figure 3A additional bacterial spots appear to have been partially eaten, or perhaps they were spotted unevenly, either way I encourage the authors to address this. Furthermore, as commented earlier, the cut-off time to determine whether an assay has reached its completion cannot be based on a “vague” 3-5 days description, this must happen consistently at an experimentally determined cut-off time point to be reproducible and avoid introduction of personal bias.
Lines 199-207: Primary screen identifies 191 mutants that are preferred feeding sources, but then the authors do not mention how many transposon mutants were identified in the second screen only stating that some from the primary screen were confirmed and others not. I ask the authors to report precisely each mutant identified in the primary screen (A), and which were identified from the secondary screen (B), highlight those that were identified from both and perhaps ignore the others. Cherry picking from each screen and composing table 2 from a combination of clones identified because they fit the hypothesis better (I.e. without explaining the basis for the choice) is not scientific reporting but biased and anecdotal.
Line 207: Please state precisely what “sufficient incubation time” would lead to the worms eating all bacteria regardless of genotype.
Figure 3B: in the cbcX plate numerous additional bacterial spots appear to have been eaten by the worms as well, please address this.
Lines 201-212: Additional explanations exist for preferred feeding behavior that are not addressed, including presence/absence of chemoattractants ( “odorants” is a vague and non-descript term) which may be soluble or cell bound, intracellular accumulation or secretion of metabolic byproducts, including quorum sensing molecules which affect “palatability by worms”, repellants – which biosynthesis of all could be affected by stage of growth, cell density or nutrient source.
Line 214: Which culture supernatants were selected for these experiments? How were culture supernatants harvested and prepared? From cultures at what density and what growth phase? Where they filter sterilized? Please list the mutants which culture supernatants were assayed.
Line 216: Please list the strains from which the culture supernatants were collected that affected the behavior of the worms which a subset of representative results in shown in figure 4, I suggest making a table for this.
Figure 4: Instead of just providing strain names please list them by name of genes and refer to them in Table 2.
Please state the rationale and significance for undertaking the experiments with the tph-1 nematodes described from line 222 onwards and what the hypothesis for this experimental set it, as this is not entirely clear why the authors would hypothesize that serotonin plays a role in detecting preferred food sources.
Line 236: Please list the 48 clones chosen for the experiment and explain the reason for the choice.
Not clear how the discussion from lines 262 to 307 helps the authors make their point.
Figure 6 is missing statistical analysis and does not strengthen the validity of the model. The strains listed there should be highlited in Table 2.
Lines 248-250: Please categorize all genes identified from the transposon mutants screen as indicated here in functional groups. This line makes clear that the authors do not have what they suggest in the title, abstract and conclusions (line 310) any “(sensitive) indicator of bacterial virulence”.

Validity of the findings

The title of the manuscript unambiguously states that feeding behavior of Caenorhabditis elegans is an indicator of Pseudomonas aeruginosa PAO1 virulence. This may be the case for T3SS mutants although the inability of the authors to provide assurances that bias is no introduced by evaluating the results of their feeding assays “between 3 and 5 days” (close to the window of time when the worms would begin to eat any bacteria regardless of genotype) is not reassuring. However, evaluation of a P. aeruginosa transposon mutagenesis library screen reveals that genes identified as preferred food sources by C. elegans can be grouped into major categories that include virulence factors, virulence regulators, hypothetical proteins and most importantly a wide range of metabolic genes. This clearly disqualifies the feeding behavior model as a as a sensitive indicator of virulence for bacterial strains that have moderate worm killing activity. In the abstract the authors claim that 37 mutants were identified as preferred food sources, yet the primary screen identified 191 mutant strains that C. elegans fed on preferentially. Table 2 lists 37 mutants that were chosen either from the primary (A) or secondary (B) screen arbitrarily or both. What was the basis for these choices? Either way, these data sets are not vigorous enough to support the idea that feeding behavior of Caenorhabditis elegans is an indicator of virulence.

Additional comments

Dear Author:
I need to pre-qualify my review by stating that I am a microbiologist with >20 years experimental experience in microbial pathogenesis and Gram negatives in particular, but I have no expertise in worm models of bacterial colonization, feeding etc. The manuscript came across to me very “foreign” and difficult to interpret. Given this, my review does not come from the perspective of someone who has intimate knowledge and understanding of this experimental model. Regardless, since the manuscript should be comprehensible to a broad audience, I do believe my review to be worthwhile and encourage you to address my comments.

---

## Round 0.2 · Minor Revisions

Thank you for your changes to the manuscript and addressing the majority of the reviewers comments. I am still not satisfied with the presentation of the data as bar graphs. While this may be common, it does not mean it is correct, and hides much of the information about where the data actually lies. Furthermore, the SEM is used to denote population variation not sample variation as you are showing. As you do not want to present the data as the individual subjects, I recommend that you present it as box-whisker plots instead.

I also strongly suggest that you add the word 'potential' in front of 'virulence factors' in your abstract and conclusions when referring to ability of assay to indentify virulence factors.

Minor corrections (track changes version):
Pg 5, line 75 - correct spelling of 'ordorants'
Pg 5, line 82 - removed first mention of 'secreted'
Pg 6, line 98 - add 'potential' before virulence factors
Pg 19, line 419 - please italicise 'Pseudomonas'
pg 20, line 432 - please add 'as' between 'approach' and 'a'

---

## Round 0.3 · accepted · Accept

Thank you for your patience during the review process.